# The Potential Role of Dopamine in Mediating Motor Function and Interpersonal Synchrony

**DOI:** 10.3390/biomedicines9040382

**Published:** 2021-04-05

**Authors:** Hila Z. Gvirts Probolovski, Anat Dahan

**Affiliations:** 1Department of Behavioral Sciences and Psychology, Ariel University, Ariel 44837, Israel; 2Ort Braude College of Engineering, Karmiel 2161002, Israel; anatdhn@braude.ac.il

**Keywords:** dopamine, predictive coding, forward model, motor planning, interpersonal synchrony

## Abstract

Motor functions in general and motor planning in particular are crucial for our ability to synchronize our movements with those of others. To date, these co-occurring functions have been studied separately, and as yet it is unclear whether they share a common biological mechanism. Here, we synthesize disparate recent findings on motor functioning and interpersonal synchrony and propose that these two functions share a common neurobiological mechanism and adhere to the same principles of predictive coding. Critically, we describe the pivotal role of the dopaminergic system in modulating these two distinct functions. We present attention deficit hyperactivity disorder (ADHD) as an example of a disorder that involves the dopaminergic system and describe deficits in motor and interpersonal synchrony. Finally, we suggest possible directions for future studies emphasizing the role of dopamine modulation as a link between social and motor functioning.

## 1. Introduction

Interpersonal synchrony relies on the alignment of behaviors in time [1], and widely occurs in natural settings, including the mother–infant relationship [2,3], walking side-by-side [4] or during conversation [5]. This ability has been linked to the mechanisms of joint attention [6] and motor coordination [7], and is also thought to play a key role in the acquisition of social cognition skills during the early years of life [8,9]. 

While it is agreed that motor functions are crucial for our ability to synchronize our movements with those of others [10], the two have largely been studied separately.

Although there is some evidence for the association between atypical intrapersonal motor functioning and atypical interpersonal motor synchrony [11,12,13], it is as yet unclear whether these two functions share common neural mechanisms. Here, we address what is arguably one of the most important questions, namely, are motor and social skills linked, and how?

The fact that both interpersonal synchrony and motor deficits have been associated with atypicalities in the function of the dopaminergic system [14,15] suggests that a framework involving the ascending dopamine system can provide a model that connects these two roles of the motor system.

In an attempt to develop such a model, we synthesize disparate recent findings on motor functioning and interpersonal synchrony and propose that these two functions share a common neurobiological mechanism and adhere to the same principles of predictive coding. Critically, we describe the pivotal role of the dopaminergic system in modulating these two distinct functions.

### 1.1. Motor Functioning and Planning

Successful motor functioning can be viewed as involving four main components [16]: (i) attention to the process; (ii) motor planning; (iii) motor execution; and (iv) motion monitoring and error correction. These components are not separate and are highly connected. When a person reaches their hand to lift up their keys, for example, they have to pay attention to the task to be performed, produce a motor plan for the movement that allows reaching and grasping the keys, execute the movement, and constantly monitor the movement to see that they reach the right location.

The process of movement planning can be identified as neural activity in the motor cortex and connected brain regions before an actual movement occurs. Recent advances in large-scale neural recordings are providing valuable insights into how the dynamics of populations of neurons are related to motor planning and movements [17,18]. The person reaching for their keys has several options (for example, reach for their phone or their keys). The process of action selection and planning is not a separate and serial process, but rather potential action plans are constantly competing [19,20] as peaks of activity in neural populations in fronto-parietal regions [21]. Several peaks that occur simultaneously within a cortical region are in competition until one peak surpasses a threshold and suppresses the others. Brain regions involved in action planning and selection include regions in the temporal lobe, such as the fusiform gyrus (FG) [22], and regions in the parietal lobe such as the dorsomedial posterior parietal cortex (dmPPC), the caudal part of the dorsal premotor cortex (PMdc) [23], the superior parietal lobule (SPL) and supra-marginal gyrus (SMG) [24], and anterior and posterior parts of the intraparietal sulcus [25], and frontal regions such as the inferior frontal gyrus (IFG) [26].

After a movement is initiated, it is constantly monitored and the actual output of the motor system and the planned outcome are compared for future motor command adjustment as part of a feedback process [27,28]. A planned outcome is described as a motor prediction. In the case of a simple movement, the connection between a motor plan and the outcome can be fairly simple. However, when we consider limb motion, the relationship between our motor plans and motor outcomes is far more complex due to the dynamics of multi-joint motion [29]. This complexity is even greater when we consider interactions that occur in real life engagement with the environment and in interactions that include tool use [22,30] which may require a further causal relationship between the tool and the goal [31,32]. For example, prediction of the consequences of a motor plan in real life complex environments requires a simulation of the dynamic behavior of our movement and environment. The prediction of the expected outcome of a motor plan taking into account the complexity of our limbs and joints, the changing environment, and possibly the tools we are using is modeled as a forward model [29,33]. If we interact with a changing environment such as a new tool or an environment with unexpected physics, our initial movement will exhibit an error, which will gradually be minimized as our brain forms a model that predicts the physics of the object or environment [34]. Interestingly, interaction with the environment involves the SMG, which is usually involved in activity within an environment, even for calculations in the context of shopping when compared to parallel arithmetic operations [35].

### 1.2. Interpersonal Synchronization

Interpersonal synchrony (IS) refers to the dynamic and reciprocal adaptation of the temporal structure of behaviors between interactive partners [1]. Interpersonal synchronization is manifested in a variety of social contexts, including the mother–infant relationship [2,3], empathic communication [36], and walking side-by-side [4] or during conversation [5]. A growing body of evidence suggests that it is an important determinant of successful social interaction [37,38,39], with interpersonal synchrony found to elicit rewarding sensations and encourage closeness and connectedness [40]. Interpersonal synchrony was also found to promote feelings of affection and belonging [41], rapport [42], trust [43], empathy [44], and collaboration [45]. Moreover, synchronization was shown to enhance the capacities for emotion regulation [46]. These studies show the importance of motor synchrony beyond the scope of learning the social world, but in actual prosocial and individual benefits. In some contexts, the ability to coordinate movements with others has a role in encounters that are not prosocial, but rather movement coordination serves a critical role in aversive encounters and in combat sports [47,48,49]. 

As opposed to mimicry and imitation, which require simple behavioral matching, synchrony requires that the movement will be performed at the same time. In other words, in order to coordinate in time, one needs to be able to understand the action of with whom they would like to synchronize and to predict when they will perform their next movement. It is therefore not surprising that synchrony has been found to foster mentalization, which is essential for efficient communication, in general, and for predicting, in particular [50]. 

As such, it has been suggested that neural synchrony, which emerges during tasks that require coordination [51,52], marks successful communication and mutual understanding [53]. Synchrony involves the timely coordination of behaviors of two or more individuals [54]; therefore, it cannot be accomplished without all the participants’ shared attention to each other’s movements. Recent evidence suggests synchronization between two or more brains during social interaction serves to increase their attention to the specific interaction and may play an important role in allocating mutual attention to those with whom we interact. It is argued that the synchronization of neural activities serves as a potential mechanism for tuning to significant others. Accordingly, neural synchrony marks the ability of all interactive partners to share their attention with one another and thereby facilitates their ability to synchronize their movements. This requires further validation, because according to the existing literature, it is impossible to accurately infer whether neural synchrony is the precursor to interpersonal synchrony or the product of it [55].

### 1.3. Predictive Coding Supports the Link Between Motor and Social Dysfunctions

Motor functioning has been repeatedly situated within the framework of predictive coding [56,57,58,59]. The predictive coding framework argues that the brain generates predictions of the consequence of a motor command in relation to the motor plan and environment with which we are interacting. Such a prediction is represented as a forward model. A forward model that generates predictions on the sensory consequences of the executed movement is considered an integral part of motor execution [60]. 

As the dynamics of our body change during development, and as we gain experiences with tools and environments which have their own intrinsic dynamics [30,31], we constantly need to acquire and update motor forward models [61,62]. The updating of forward models is achieved using prediction errors, by comparing the predicted and actual outcome of a motor command. Processes of learning translate prediction errors into changes in synaptic weights which will improve future predictions of the forward model.

It is plausible to assume that as the forward model models the dynamics of a motor plan outcome in interaction with a dynamic environment, it will also model the outcome in relation to a partner in the case of social alignment. Indeed, it was argued that in an attempt to conserve computational resources, the brain tends to optimize the representation of the environment, which includes those with whom we interact, their brains, and their behavior [63]. It was proposed that “behavioral synchronization observed in interacting human agents is a consequence of individual brains in interaction with each other operating under a general optimization principle” [36] (p. 7). To test whether interactive partners need to adapt their own predictive models in order to synchronize with one another, Heggli et al. [64] designed a paradigm in which a dyad of musicians were asked to synchronize their movements under two conditions: in the shared model condition, they were asked to tap while listening to the same rhythm, and hence had the same top-down predictive mode, while in the non-shared model condition, they were asked to tap to a different rhythm (one participant to polyrhythm and the other to a straight rhythm), and hence had two conflicting predictive models. Their findings revealed that tapping synchronization was significantly worse at the start of trials with non-shared models compared to trials with a shared model, suggesting that different predictive models impact synchronization in musicians performing joint finger tapping. 

Surprisingly, despite accumulating evidence suggesting that the brain can best be understood within a predictive coding framework [65,66,67], it was only recently that the mechanism of interpersonal synchronization was considered within this framework. Shamy-Tsoory et al. [40] described the neural mechanism that underlies our ability to synchronize with others as an error monitoring system. According to this framework, misalignment between our movements and the movements of others triggers prediction error and induces distress, whereas achieving an alignment with others activates the reward system and reduces distress. Gebauer et al. [68] suggested the role of online prediction, based on past events and adaptive error correction, leading to continuous motor adjustment in synchronization. Koban et al. [63] suggested predictive coding and the free energy principle as possible explanations for interpersonal coordination dynamics, pointing out that individual brains attempt to reduce the mismatch between observed and their own motor behavior. Era et al. [69] also commented that models relating interpersonal interaction should consider the involvement of the error monitoring system. 

## 2. Dopamine and the Neural Mechanism That Governs Error Detection

In what follows, we attempt to review evidence in support of the notion that the same predictive coding principle operates across both motor functioning and social cognition. While it is reasonable to suggest that atypical predictive coding may lead to atypical motor functioning, and, thereby, to reduced alignment with an interaction partner, this possibility has so far received little research attention. However, it is also possible that less weighting of the social context or history of the interaction may also generate a prediction error which could also lead to reduced alignment with an interaction partner. Critically, given that dopamine neurons in the midbrain signal a reward prediction error (i.e., differences between received and predicted rewards) [70], we suggest that the dopaminergic system plays a central role in regulating the ongoing processing of error detection and error correction. Finally, we advocate in favor of a common set of brain networks underlying motor functioning and interpersonal synchrony. 

Monitoring of motor functioning requires the detection of differences in expected and actual outcomes. Errors are monitored and detected in a feedback control loop on several levels: within a trial [71], in the following trial [72,73], and in the span of several trials [73,74,75]. Several studies have consistently shown the involvement of the medial frontal cortex (MFC) and, specifically, the anterior cingulate cortex (ACC) [76,77], in error detection during motor functioning. ACC activation mediates motor functioning and was shown to be activated in the detection of mismatch between desired and actual outcomes of movement [27]. Moreover, underactivity in these regions has been associated with deficits in error detection and subsequent adjustments in participants with several disorders such as attention deficit hyperactivity disorder (ADHD), schizophrenia, and substance abuse [78,79,80].

These areas seem to be involved in monitoring and error detection in other social and non-social contexts [81]. The ACC was found to be mainly sensitive to unexpected outcomes or the unpredictability of the outcomes [82]. The medial frontal cortex and, in particular, the ACC, are mediated by dopamine via corticobasal ganglia circuits [83,84]. Importantly, dopamine response has been suggested to reflect a reward prediction error that occurs when prediction does not match with actual experienced outcomes [70]. 

The reinforcement learning theory [82,85] posits that the medial prefrontal/ACC receives reward prediction error signals from midbrain dopaminergic cells, which also send the same input to the basal ganglia. Accordingly, when the error signal is delivered to the ACC, it contributes to changes in attentional focus to enable the brain to adapt behavior to changing task demands and environmental circumstances [27]. In addition, activation of the ACC seems to contribute to a choice between several competing possible actions [86,87]. It was therefore proposed that the ACC plays a crucial role in the hierarchical mechanism for action selection during motor planning and thus acts as a “motor control filter” to decide which motor policy is the most suitable for each specific task [82,85]. Notably, the value of each possible action is learned via signals carried to the ACC by the midbrain dopamine system [81]. 

In line with the idea that motor functioning and interpersonal synchrony are mediated by common neural mechanisms that govern error detection, activity in the ACC and dorsomedial prefrontal cortex (dmPFC) was found in studies on interpersonal synchrony [40]. More specifically, activation in these regions has been argued to underlie the ability to estimate the amount of distance (in movement) between the individual and interactive partner/s and to detect misalignment. It should be noted that while the dorsal part of the ACC (dACC) was found to be sensitive to expectancy violation, the ventral part of the ACC (vACC) was found to be sensitive to social distress and social rejection [88]. Indeed, it has been proposed that the dACC is more involved in error monitoring than distress, and thus may be more relevant to our model. Notably, this region seems to play a major role in comparing the value of possible actions and determining what would be more beneficial: maintaining the same behavior or changing it [89,90,91,92]. Given the involvement of dopamine in planning our motor actions by modulating the value of each possible action, it is reasonable to suggest that it may also determine the value of possible actions when the goal is to align our movements with others. In support of this notion, Gvirts and Perlmutter [89] (p. 5) proposed that dopamine not only “facilitates the ability of tuning-in to a specific social interaction while tuning out other interactions, but actually regulates the reward value of each of the potential interacting partners, thereby prioritizing and determining the amount of attention that should be allocated to each of them”. However, so far, the role of the dopaminergic system in updating the social values associated with the interaction with a specific partner/s is yet to be determined. 

Accumulating evidence indicates that the process of motor error correction triggers the activation of the observation–execution (OE) system. The OE (also often referred to as the mirror neuron system (MNS)), describes a system of brain regions that demonstrate similar responses during both the observation and execution of actions [93]. These regions include the inferior frontal, premotor, and inferior parietal regions [94,95,96].

For example, the inferior frontal gyrus (IFG), which has been implicated in the representation as well as in the preparation of action (26), was shown by several studies to be involved in processes of error correction [97,98]. The pre-SMA and the IFG were also shown to be involved in stop signal task performance [99,100]. The role that these fronto-temporal regions play in higher order motor control is interpreted as a proactive control system that allows inter-trial adjustments to the level of motor readiness based on prior performance and anticipated task requirements [101,102,103].

Interestingly, the role of the OE system in reading or recognizing the goals of observed actions has been suggested to be understood within a predictive coding framework. Accordingly, action understanding involves a process of minimizing the prediction error between a forward model of the most likely cause of an input and the actual input. In line with the predictive account, action understanding could rely on internal forward models to anticipate the unfolding of a given action [60]. 

As noted above, according to the prediction model framework, when the behavior of others is matched with our own, the environment is more predictable, prediction error is minimized, and it is easier to reason about their mental states [104]. Interpersonal synchrony requires the ability to anticipate the movement of the interactive partner so that the movement can be matched in time; therefore, it is not surprising that interpersonal synchrony has been shown to involve the activation of the OE system [105]. In particular, the inferior frontal gyrus (IFG), a core region of the OE system, was found to be critically involved in the neural network underlying interpersonal synchrony (e.g., mimicry [106] and imitation [107] and, more recently, the IFG was found to be activated during interpersonal synchrony [108,109]). Furthermore, evidence from hyperscanning studies suggests that coupling in the IFG serves as the neural mechanism that mediates interpersonal synchrony [52,110,111]. 

Taken together, this line of evidence supports the notion that motor functioning and interpersonal synchrony are mediated by a common neural mechanism governing error correction. As noted above, the underlying mechanism of interpersonal synchrony, as suggested by Shamay-Tsoory and others (2019), is that the activation of the reward system leads to the rewarding sensations associated with interpersonal synchrony. More recently, Gvirts and Perlmutter [112] have proposed that the activation of the reward system triggers the release of dopamine which potentially activates what they refer to as the mutual social attention system of interacting partners (i.e., the coupling between participants’ temporoparietal junctions and/or prefrontal cortices). Gvirts and Perlmutter suggest that the role of the dopaminergic system is to regulate the reward value of each of the potential interacting partners, thereby determining with whom we synchronize. Indeed, interactions between the mesolimbic dopamine system have been shown to direct attention and assign salience to relevant information depending on its valence (e.g., positive or negative information) [113,114]. Hence, we propose that while interpersonal synchrony triggers rising levels of dopamine, this, in turn, may lead to an increase in social salience of both positive and negative social cues. The crucial role of dopamine may therefore explain why the effects of interpersonal synchrony are not always positive and may be context-dependent. As such, synchronous activity was found to boost harmful behavior such as compliance with requests to exhibit aggression and destructive obedience [115]. Likewise, in non-human species, coordinating movements with another animal is also critical in competitive and aversive encounters [47]. 

Thus, we suggest a shared neural mechanism for interpersonal synchronization and motor functioning, using a mechanism of error detection by comparing outcomes to a predictive forward model. Moreover, we suggest that when synchronization occurs, interactive partners share a common predictive model that governs the movement plan of each interactive partner and facilitates a constant updating of the forthcoming actions of others.

The process of motion planning and error correction for a single person performing a movement can be described as follows. A person receives sensory input regarding their own location and the state of the environment. Then, the individual generates a motor plan that is modulated by dopaminergic projections according to the state of the environment. A forward model of the predicted outcome of the movement, taking together the movement plan and the environment, is generated. As the movement is executed, it is compared to the forward model. Accordingly, errors in the motor plan and in the forward model are detected. The processes of movement selection and of movement correction are modulated by dopaminergic projections from the striatum. When a prediction error occurs (i.e., when rewards differ from their prediction), dopamine neurons respond. Subsequently, the movement plan and the forward model are updated (Figure 1).

The process of motion planning and error correction for a person interacting in joint movement with a partner can be described as follows. A person receives sensory input regarding their own location and/or the location and movement of an interacting partner. A prediction regarding the movement of the other and a motor plan to align with that movement is then generated. The motor plan is modulated by dopaminergic projections according to sensory influences and a shared predictive model of the shared movement. Accordingly, a forward model of the expected outcome related to the movement plan and the movement of the partner is generated. Both the person and the other interactant continue their movement. A similar process occurs for the partner (who also predicts and aligns their movement according to a shared predictive model).

Errors in the executed movement are compared to the existing forward model that is based on a shared model of self and other and the movement plan. When a prediction error occurs—that is, when rewards differ from their prediction—dopamine neurons respond. 

Accordingly, the shared predictive model for the person and the partner is updated and the movement plan and the predicted forward model are updated. The process of movement correction is modulated by dopaminergic projections from the striatum. 

Hence, an ongoing process of motor synchronization involves constant cycles of updating the motor plan and creating a forward model, detecting error between the desired outcome and the actual movement of our own movements and our partner’s movements, and further updating the motor plan and forward model. This process in all its stages is modulated by dopamine projection in the selection of a movement plan, in signaling of an error, and in choosing how to update the motor plan (Figure 2).

## 3. Motor and Social Deficiencies in Developmental Disorders That Involve the Dopaminergic System 

Some neurodevelopmental disorders involve difficulties in both motor functioning and social cognition. For example, individuals with Parkinson’s disease—which is primarily associated with motor functioning deficits—show social cognition deficits [116], while people with autism spectrum disorder (ASD)—which is primarily associated with social cognition deficits—show motor functioning deficits [117]. It is increasingly acknowledged that motor and social skills may be linked [118], but how? Some evidence, mostly anecdotal at this stage, suggests that interpersonal synchrony may form the “missing link” between the two. For instance, it has been repeatedly demonstrated that individuals with ASD exhibit, in addition to the hallmark social cognition deficits, aberrant performance in tasks involving interpersonal synchrony [12,119], which recent studies suggest may be, at least partly, accounted for by motor control deficits in this population [120]. 

Here, we present attention deficit hyperactivity disorder (ADHD), as an example of a disorder that involves the dopaminergic system, and review findings regarding both motor and social difficulties and recent findings regarding difficulties in interpersonal synchronization. Finally, we put forward evidence linking dopaminergic modulation by the administration of methylphenidate to improvements in both motor and social domains.

Individuals with ADHD exhibit motor difficulties, both in fine motor skills [121,122,123] and in gross ones [124,125], which appear even during the first years of life [126,127]. A variety of clinical tests have shown that ADHD involves motor dysfunction [16,128]. For example, deficits in motor planning have been shown in ADHD, by comparing reaching movements after sufficient (long) and insufficient (short) planning intervals. It was found that when sufficient planning time is provided, participants without ADHD seem to have a motor plan ready, whereas those with ADHD do not initiate the movement immediately after the cue even when provided with a long planning interval, possibly indicating that they do not have a motor plan ready [16].

It has been demonstrated that children with ADHD display inadequate social behavior [129] and suffer from social problems in reciprocal relationships [130,131,132]. Notably, these symptoms tend to persist over the lifespan of the disorder, because adults with ADHD display friendship problems, poorer social interactions [133,134], loneliness [135], poorer intimate relationships and marital adjustments [136], which collectively reflects their difficulties in social interaction. Social cognition deficits can have substantial consequences on the quality of social functioning [137,138]; therefore, it is surprising that studies of social cognition deficits in adults with ADHD are relatively scarce. Recent studies [139,140,141] have shown impairments in empathy and theory of mind (ToM) functioning in children with ADHD. It has been suggested that, while impairments in empathy may persist over time, ToM functioning normalizes in adulthood in people with ADHD [142,143]. 

Notably, reduced interpersonal synchronization appears to be associated with deficits in social cognition, e.g., in schizophrenia [144,145] and in autism spectrum disorder (ASD) [119,146], where interpersonal synchronization was suggested to be significantly associated with social skills [11,40,44]. Indeed, developmentally, interpersonal synchronization has been found to play an important role in the acquisition of social cognition skills [8,9]. In ADHD, it has been shown that the extent of interpersonal synchronization between a mother and her child is associated with the degree of functioning of preschool children with ADHD [147]. Recently, Gvirts et al. showed for the first time that ADHD is associated with deficits in intentional synchrony in tasks requiring a pair of participants to move their hand together [148].

As discussed above, both social and motor deficits have been associated with atypicalities in the function of the dopaminergic system. In ADHD, the effect of dopaminergic modulation on social cognition and motor functions has been studied separately. In the social domain, dopaminergic simulants were found to reduce negative social interactions and to improve social and behavioral functioning in children with ADHD [149,150], and it has been shown that a single dose of methylphenidate (MPH) improves the performance of children with ADHD in ToM-related tasks [151]. This is in line with the reported improvement in the empathy scores of children with ADHD following an MPH treatment [152]. A recent study found that after social interaction (i.e., a parent–child interaction), oxytocin levels were significantly higher in the healthy control group compared to children with ADHD. Importantly, the administration of MPH attenuated this difference such that after social interaction, differences in oxytocin levels between children with ADHD and healthy controls were no longer found. These findings shed light on a possible mechanism by which stimulants improve social abilities in ADHD. The study raises the possibility that oxytocin plays a role in mediating the improvement in social cognition among children with ADHD who are treated with stimulants.

In the motor domain, dopaminergic stimulants administered to children with ADHD have been shown to improve some motor functions, e.g., dynamic balance [153,154], coordination [155] and fine and gross motor skills [156], but not others (e.g., static-balance) [157]. 

## 4. Discussion

Predictive coding views the brain as a “prediction machine” [158]. For example, in the visual system, the objects that most likely give rise to current retinal input are allowed to be inferred [159], as well as in problems of actions determining which action is more likely to generate preferred outcomes [159], and in motor functioning [58]. The role of dopamine in this model has been highlighted in numerous studies [160], linking dopamine to a reward prediction error (RPE)—the discrepancy between observed and expected reward [70]. Recent evidence suggests that dopamine also signals the amount of surprise associated with a rewarding outcome or stimulus [161].

Here, we discussed the possibility that motor functioning and interpersonal synchrony have shared common neurobiological mechanisms and that both functions adhere to principles of predictive coding. 

Although interpersonal synchronization was mostly shown to play an important part in positive interactions in humans and also for non-human species [162,163], coordinating movements with others is also critical in competitive aversive encounters and in combat sports [47,48,49]. By articulating the role of dopamine in increasing social salience, our model sheds light on the link between interpersonal synchrony and negative social interactions. The model, therefore, complements the existing literature by suggesting a neurobiological model that is not restricted to positive social interactions and can be applied broadly across all kinds of social interaction. It is important to note, however, that because this is a novel model, it requires validation, and here we would like to emphasize the importance of investigating the proposed role of dopamine in the form of increased intake (e.g., with methylphenidate).

It should be noted that although our proposed model provides a framework linking motor planning and interpersonal synchrony, it does not provide a complete explanation for all behavioral aspects of interpersonal synchronization. For example, according to the predictive coding theory, once interpersonal synchronization is achieved, the expected outcome and actual outcome are the same and the midbrain dopamine neurons will not signal an error. In this case, why would interacting partners not continue indefinitely with this form of movement? Given that predictive coding does not account for the inherent role of variability in behavior [47], our model does not provide an explanation for the alterations between moving in and out of synchrony that characterize social interactions [48,49,164]. While moving in synchrony induces a strong sense of connectedness with others and a feeling of self–other merging, moving out of synchrony allows for self–other distinction [165], which is crucial for the process of understanding others’ mental and emotional states [166]. This is especially relevant for rapid interpersonal interactions which create continuous disturbances and rapid alterations between moving in and out of synchrony [47,48]. 

The alterations between moving in and out of synchrony may be explained in view of perceptual control theory (PCT) [47,167]. According to this theory, goal states are achieved by negative feedback control mechanisms, as in a mechanism governing homeostatic systems. As noted by Bell et al., when considering two events of switching a thermostat on and off, “the two events are not unrelated, and in fact, are intrinsically and continuously interconnected” (p. 342). The feedback is used here to change the behavior itself and this change is continuous—the behavior is changed, and the feedback continuously modifies the behavior in an infinite cycle. Similarly, our tendency to move in and out of synchrony with others may be seen as two events that are not unrelated: two partners that achieve interpersonal synchronization will after a certain time misalign and move out of synchrony as a means of controlling some higher-level perceptual variables (e.g., self–other distinction). The feedback will continue to modify their behavior, such that after a certain time they will move into synchrony again. This idea is compatible with findings by Feniger-Schaal et al. [164] in which the association between synchrony and attachment style has been tested. Secure attachment is the result of appropriately responsive parenting and was found to predict the development of higher social skills [168] than insecure-dismissing attachment. Interpersonal synchrony is thought to play a key role in the acquisition of social cognition skills during the early years of life [8,9]; therefore, different attachment styles were expected to be associated with different patterns of interpersonal synchrony. To examine the link between attachment style and interpersonal synchrony, a mirror game paradigm has been employed, which allows the quantification of the synchrony between two players [169]. Performance on this task was compared between pairs with secure attachment and pairs with insecure attachment. Interestingly, the findings revealed that security of attachment was related to a more exploratory and less rigid game than insecure-dismissing attachment. In other words, pairs with secure attachment allowed themselves more freedom to explore during the game, and displayed a larger variety of movement patterns. Future research will be necessary to provide a framework that connects motor function and our tendency to move in and out of synchrony with others. Such a model will allow us to examine whether disorders that involve the dopaminergic system are characterized by deficiencies in the capacity to explore and to display a variety of movement patterns. 

Taken together, we propose a novel model that may help to elucidate our understanding of the role that dopamine plays in linking motor planning and interpersonal synchrony, and thus may potentially hold important theoretical and clinical implications.

This framework can, therefore, guide detailed investigations of whether deficits in interpersonal synchrony are due to motor function (i.e., shared neural mechanisms) or additional factors beyond motor functions (e.g., the reward value associated with a specific partner/social interaction). This framework may also bridge the gap between two distinct fields in the literature: the field of interpersonal synchrony and that of motor function. Finally, this framework can potentially be tested in other disorders in which both social and motor skills are impaired. As noted above, we call for future studies to investigate the proposed role of dopamine within the suggested model by methods of increasing dopamine (e.g., by methylphenidate intake) in a variety of contextual social cues.

## Figures and Tables

**Figure 1 biomedicines-09-00382-f001:**
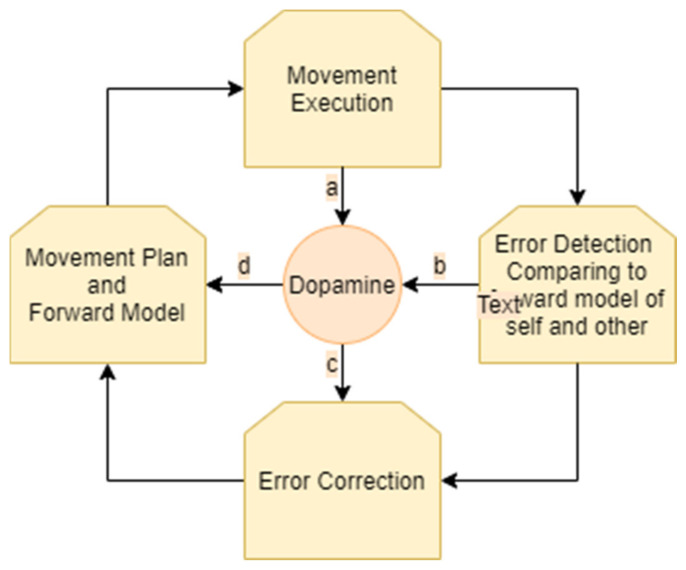
Monitoring and correction of movement: an ongoing cycle involves generation of a movement plan and forward model, movement execution, error detection, and error correction. Dopamine is involved in the modulation of these stages. Dopamine influences the choice of the movement plan (**a**). As the movement is executed, it is constantly monitored and compared to the forward model, resulting in prediction error. This process of monitoring and error detection influences the levels of dopamine (**b**,**c**). Dopamine influences error correction (**d**) which results in a new movement plan and forward model.

**Figure 2 biomedicines-09-00382-f002:**
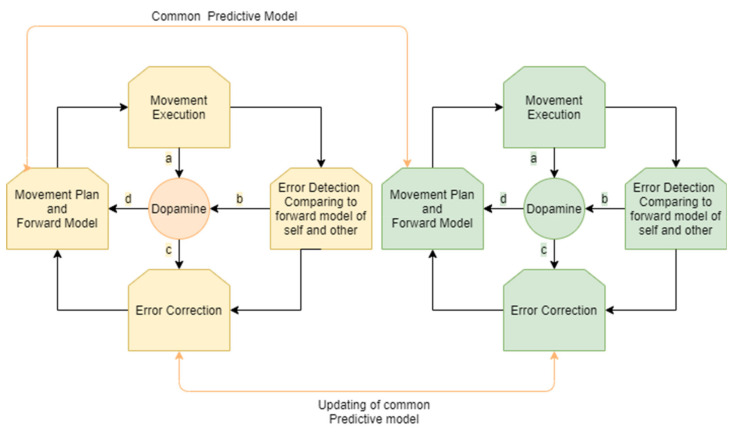
Monitoring and correction of interpersonal movement synchronization: an ongoing cycle for both participants involves the generation of a movement plan and forward model according to a shared predictive model, movement execution, error detection, and error correction. Dopamine is involved in the modulation of these stages, influencing the choice of the movement plan with a common predictive model (**a**). As the movement is executed, the movement of the player and their partner is constantly monitored and compared to the forward model, resulting in prediction error in cases of misalignment. This process of monitoring and error detection influences the levels of dopamine (**b**,**c**). Dopamine influences error correction (**d**) that results in a new movement plan and forward model.

## Data Availability

Not applicable.

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
