# Peer review of "The Potential Role of Dopamine in Mediating Motor Function and Interpersonal Synchrony"

_biomedicines, 2021, doi:10.3390/biomedicines9040382_

Round 1

Reviewer 1 Report

As the authors point out, even though motor planning is crucial for both individually oriented tasks and social interactions, the two have largely been studied separately. The present paper develops a framework involving the ascending DA system and its connections that connects these two roles of the motor system. Importantly, the paper integrates much of the existing empirical literature and the model is sufficiently well developed to stimulate explicit empirical tests of its core elements. I have no substantive concerns with the paper, but there are a couple of issues that are either unclear or not adequately explained. There is also one minor stylistic change that I recommend.

Interpersonal synchrony is discussed with regard to positive affective states, and as made clear from the human literature in the present review, such synchrony has also been shown to be important for non-human species in facilitating the prolongation of positive interactions (e.g., Palagi, E. et al. (2020). Mirror replication of sexual facial expressions increases the success of sexual contacts in bonobos. Scientific Reports, 10, 18979; Scopa, C., & Palagi, E. (2016). Mimic me while playing! Social tolerance and rapid facial mimicry in macaques (Macaca tonkeana and Macaca fuscata). Journal of Comparative Psychology, 130, 153–161). However, coordinating movements with another animal is also critical in competitive, aversive, encounters, and this can be highly fine-tuned (Bell, H. C. (2014). Behavioral variability in the service of constancy. International Journal of

Comparative Psychology, 27, 196-217). The impression gained from the present paper is that the model that is developed is restricted to positive social interactions. Is this a misimpression of the model? If so, make clear that it applies broadly across all kinds of social interactions. If not, then provide a rationale for why competitive and agonistic social interactions belong in a class of their own.

Second, rapid interpersonal interactions create near continuous disturbances that need correcting (e.g., Blanchard, R. J., Blanchard, D. C., Takahashi, T., & Kelley, M. J. (1977). Attack and defensive behaviour in the albino rat. Animal Behaviour, 25, 622–634). In the model used in the present paper, the correction is for predicted misalignment of movements, but surely misalignment of controlled perceptions would be quicker and more economical to execute (e.g., see Bell, 2014 cited above). A discussion of these alternatives and a rationale for the type of misalignment used in the present model should be provided. This is especially problematic since it is the failure to achieve the predicted outcome that provides the feedback that blocks dopamine reward. It is unclear how such a mechanism could provide the rapid corrections to movements that are evident in dynamic interactions.

Line 90: Change “Forward model that generates predictions…” to either “A forward model that generates predictions…” or “Forward models that generate predictions…”

Author Response

Reviewer: 1

As the authors point out, even though motor planning is crucial for both individually oriented tasks and social interactions, the two have largely been studied separately. The present paper develops a framework involving the ascending DA system and its connections that connects these two roles of the motor system. Importantly, the paper integrates much of the existing empirical literature and the model is sufficiently well developed to stimulate explicit empirical tests of its core elements. I have no substantive concerns with the paper, but there are a couple of issues that are either unclear or not adequately explained. There is also one minor stylistic change that I recommend.

*Interpersonal synchrony is discussed with regard to positive affective states, and as made clear from the human literature in the present review, such synchrony has also been shown to be important for non-human species in facilitating the prolongation of positive interactions (e.g., Palagi, E. et al. (2020). Mirror replication of sexual facial expressions increases the success of sexual contacts in bonobos. Scientific Reports, 10, 18979; Scopa, C., & Palagi, E. (2016). Mimic me while playing! Social tolerance and rapid facial mimicry in macaques (Macaca tonkeana and Macaca fuscata). Journal of Comparative Psychology, 130, 153–161). However, coordinating movements with another animal is also critical in competitive, aversive, encounters, and this can be highly fine-tuned (Bell, H. C. (2014). Behavioral variability in the service of constancy. International Journal of Comparative Psychology, 27, 196-217). The impression gained from the present paper is that the model that is developed is restricted to positive social interactions. Is this a misimpression of the model? If so, make clear that it applies broadly across all kinds of social interactions. If not, then provide a rationale for why competitive and agonistic social interactions belong in a class of their own.

Response:

We thank the reviewer for this constructive comment. As suggested, we now discuss in more depth our model and how it can be applied to all kinds of social interactions.

This point is now specifically addressed in the revised Introduction section (lines: 255-264):

“Indeed, interactions between the mesolimbic dopamine system have been shown to direct attention and assign salience to relevant information depending on its valence (e.g., positive or negative information (113,114)). Hence, we propose here that while interpersonal synchrony triggers rising levels of dopamine, this, in turn, may lead to an increase in social salience of both positive and negative social cues. The crucial role of dopamine may therefore explain why the effects of interpersonal synchrony are not always positive and may be context-dependent. As such, synchronous activity was found to boost harmful behavior such as compliance with requests to exhibit aggression and destructive obedience (115). Likewise in non-human species, coordinating movements with another animal is also critical in competitive and aversive encounters (47).”    

And in the revised Discussion section (lines: 397-401):

“By articulating the role of dopamine in increasing social salience, our model sheds light on the link between interpersonal synchrony and negative social interactions. The model, therefore, complements the existing literature by suggesting a neurobiological model that is not restricted to positive social interactions and can be applied broadly across all kinds of social interaction.”

Second, rapid interpersonal interactions create near continuous disturbances that need correcting (e.g., Blanchard, R. J., Blanchard, D. C., Takahashi, T., & Kelley, M. J. (1977). Attack and defensive behaviour in the albino rat. Animal Behaviour, 25, 622–634). In the model used in the present paper, the correction is for predicted misalignment of movements, but surely c(e.g., see Bell, 2014 cited above). A discussion of tc. This is especially problematic since it is the failure to achieve the predicted outcome that provides the feedback that blocks dopamine reward. It is unclear how such a mechanism could provide the rapid corrections to movements that are evident in dynamic interactions.

Response:

As suggested, we now note that the model detailed here needs to be considered with caution as this model does not provide a complete explanation for all behavioral aspects of interpersonal synchronization. The following paragraph was added to the revised Discussion (lines: 406-418):

“It should be noted that although our proposed model provides a framework linking motor planning and interpersonal synchrony, it does not provide a complete explanation for all behavioral aspects of interpersonal synchronization. For example, according to the predictive coding theory, once interpersonal synchronization is achieved, the expected outcome and actual outcome are the same and the midbrain dopamine neurons will not signal an error. In this case, why would interacting partners not continue indefinitely with this form of movement? Given that predictive coding does not account for the inherent role of variability in behavior (47), our model does not provide an explanation for the alterations between moving in and out of synchrony that characterize social interactions (48,49,165). While moving in synchrony induces a strong sense of connectedness with others and a feeling of self-other merging, moving out of synchrony allows for self–other distinction (166), which is crucial for the process of understanding others' mental and emotional states (167). This is especially relevant to rapid interpersonal interactions which create continuous disturbances and rapid alterations between moving in and out of synchrony (47,48).”

Following the reviewer’s comment, we now elaborate on the Perceptual Control Theory (PCT) and how this theory might help to explain our tendency to move in and out of synchrony with others (lines 419-446)

*  Line 90: Change “Forward model that generates predictions…” to either “A forward model that generates predictions…” or “Forward models that generate predictions…”

Response:

As suggested, this sentence was revised.

Reviewer 2 Report

Dear Authors,

This manuscript deals with an interesting topic that has not yet been well covered in the literature. However, in order for it to be published it requires a major revision. See my comments pasted below.

  1. Keywords contain words appearing in the title (e.g., Dopamine, motor function, interpersonal synchrony) which is incorrect as it lowers the quality of searching articles in data research. Please indicate keywords that do not appear in the title.
  2. The introduction is basically the first part of the abstract word for word. It does not show a clear problem and hypothesis, which was attempted to be unraveled through this analysis. The introduction should be expanded by showing a clear theoretical problem and a clear hypothesis, but also the importance of this work. All the text up to line 131 should be the introduction, and the text on lines 121-131 should be reformulated to clearly show the problem and the hypothesis.
  3. The manuscript is not well written, many phrases do not meet the standards of scientific work, e.g., in the sentence "When a person reaches his hand to lift up his keys he would have to pay attention to the task he has to preform, to produce a motor plan for the movement that would allow him to reach and grasp the keys, execute the movement, and constantly monitor the movement to see that he reaches the right location" (lines 33-35), the authors use the phrases "his" and "he" instead of "his/her", "he or she". This is just one example (see line 40 and further; but cf. line 72). So, I have considerable ethical doubts here.
  4. When the Authors talk about planning activities (line 42-45) they refer to very old literature. I do not understand why only the IFG is referenced here. SMG, and other parietal areas, are undoubtedly the key to hand movement planning, as indicated by the latest research, see e.g.:
    • Decoding brain states for planning functional grasps of tools: a functional magnetic resonance imaging multivoxel pattern analysis study. Journal of the International Neuropsychological Society24(10), 2018, 1013-1025.
    • Effector selection precedes reach planning in the dorsal parietofrontal cortex. Journal of Neurophysiology, 108(1), 2021, 57–68.
    • The neural correlates of planning and executing actual tool use. The Journal of Neuroscience, 34(39), 2014, 13183–13194.
    • The temporal involvement of the left supramarginal gyrus in planning functional grasps: A neuronavigated TMS study. Cortex111, 2019, 16-34.
  5. For the thesis: "This complexity is even greater when we consider interactions that occur in real life engagement with the environment and in interactions that include tool use. Prediction of the consequences of a motor plan in real life complex environments requires a simulation of the dynamic behavior of our movement and environment." (line 52-54) please provide references, also in the context of ecological relevance, e.g.:
    • The neural basis of human tool use. Frontiers in Psychology5:310 2014.
    • The neural underpinnings of haptically guided functional grasping of tools: an fMRI study. NeuroImage194, 2019, 149-162.
    • Mental shopping calculations: A transcranial magnetic stimulation study. Frontiers in Psychology11:1930, 2020.
  6. The statement "In accordance with this, it has been suggested that when we move together in time with others, our minds become tuned to their minds (lines 76-77)) sounds very unscientific. Are the Authors sure that they are introduced to the philosophical category of mind, they have justification here? Is it not about the synchronization of neural mechanisms through their mutual interaction? Lines 81 and 84 indicate that the Authors mean just that. So why such a conceptual mess?
  7. Entire paragraphs appear in the manuscript without reference, although the Authors reconstruct the debate, e.g., lines 93-99. This is unacceptable in the review article.
  8. Since in the paragraph "Dopamine and the neural mechanism that govern Error Detection", the authors initially introduce data on dACC, why they do not show to what part of ACC the subsequent results concern?
  9. Figure 1 (and beyond) is of very low quality and is prepared in a non-professional program, possibly a basic shape from Word. Review articles must have professional figures, I cannot imagine publishing this work with such infantile figures. Please introduce panels to all figures, prepare them in high resolution and add extensive figure captions.
  10. Why do the Authors use capital letters for "Social" on line 257
  11. Considerations do not end with a wise, decent discussion, compilation, forecasts and interpretations. Why is there no Discussion? This work requires a thorough mind and elaboration of a deep discussion, with reference to the most recent works. In the works cited in the manuscript, I do not find a single one from 2020, and most of them are over 10 years old.

Sincerely,

Author Response

Reviewer: 2

This manuscript deals with an interesting topic that has not yet been well covered in the literature. However, in order for it to be published it requires a major revision. See my comments pasted below.

*Keywords contain words appearing in the title (e.g., Dopamine, motor function, interpersonal synchrony) which is incorrect as it lowers the quality of searching articles in data research. Please indicate keywords that do not appear in the title.

As suggested by the reviewer, we now indicate keywords that do not appear in the title.

*The introduction is basically the first part of the abstract word for word. It does not show a clear problem and hypothesis, which was attempted to be unraveled through this analysis. The introduction should be expanded by showing a clear theoretical problem and a clear hypothesis, but also the importance of this work. All the text up to line 131 should be the introduction, and the text on lines 121-131 should be reformulated to clearly show the problem and the hypothesis

Response:

As suggested, we have now added a short section to our introduction, regarding the problem and the hypothesis that led to the development of our suggested model. We also divided the rest of the introduction into three parts: the first and the second parts review the two distinct fields in the literature – the field of Motor Functioning and Planning and the field of Interpersonal Synchronization. The third part presents evidence in support of the notion that these two functions are situated within the framework of predictive coding.

*The manuscript is not well written, many phrases do not meet the standards of scientific work, e.g., in the sentence "When a person reaches his hand to lift up his keys he would have to pay attention to the task he has to preform, to produce a motor plan for the movement that would allow him to reach and grasp the keys, execute the movement, and constantly monitor the movement to see that he reaches the right location" (lines 33-35), the authors use the phrases "his" and "he" instead of "his/her", "he or she". This is just one example (see line 40 and further; but cf. line 72). So, I have considerable ethical doubts here.

Response:

Thank you for this important comment. We have sent the manuscript for additional English editing.

*When the Authors talk about planning activities (line 42-45) they refer to very old literature. I do not understand why only the IFG is referenced here. SMG, and other parietal areas, are undoubtedly the key to hand movement planning, as indicated by the latest research, see e.g.:

  • Decoding brain states for planning functional grasps of tools: a functional magnetic resonance imaging multivoxel pattern analysis study. Journal of the International Neuropsychological Society, 24(10), 2018, 1013-1025.
  • Effector selection precedes reach planning in the dorsal parietofrontal cortex. Journal of Neurophysiology, 108(1), 2021, 57–68.
  • The neural correlates of planning and executing actual tool use. The Journal of Neuroscience, 34(39), 2014, 13183–13194.
  • The temporal involvement of the left supramarginal gyrus in planning functional grasps: A neuronavigated TMS study. Cortex, 111, 2019, 16-34.

Response:

We thank the reviewer for the fascinating references. We have accordingly integrated these articles into our manuscript, utilizing them to better fine-tune and elaborate on the neural mechanisms that govern motor planning.

For the thesis: "This complexity is even greater when we consider interactions that occur in real life engagement with the environment and in interactions that include tool use. Prediction of the consequences of a motor plan in real life complex environments requires a simulation of the dynamic behavior of our movement and environment." (line 52-54) please provide references, also in the context of ecological relevance, e.g.:

  • The neural basis of human tool use. Frontiers in Psychology, 5:310 2014.
  • The neural underpinnings of haptically guided functional grasping of tools: an fMRI study. NeuroImage, 194, 2019, 149-162.
  • Mental shopping calculations: A transcranial magnetic stimulation study. Frontiers in Psychology, 11:1930, 2020.

Response:

We thank the reviewer for these important and interesting references. We added the references that you recommended and some additional references regarding environment and tool use and predictive coding.

*The statement "In accordance with this, it has been suggested that when we move together in time with others, our minds become tuned to their minds (lines 76-77) sounds very unscientific. Are the Authors sure that they are introduced to the philosophical category of mind, they have justification here? Is it not about the synchronization of neural mechanisms through their mutual interaction? Lines 81 and 84 indicate that the Authors mean just that. So why such a conceptual mess?

Response:

Following the reviewer’s remark, this sentence was revised and we now state that neural synchrony emerges during tasks that require coordination (Hu et al., 2017; Osaka et al., 2015) and this is thought to underlie successful communication and mutual understanding (Nummenmaa et al., 2018).

The revised Introduction addresses this isuue (lines 106-117):

“As such,, it has been suggested that neural synchrony which emerges during tasks that require coordination (51,52), marks successful communication and mutual understanding (53). Since synchrony involves the timely coordination of behaviors of two or more individuals (54), it cannot be accomplished without all the participants’ shared attention to each other’s movements. Recent evidence suggests synchronization between two or more brains during social interaction serves to increase their attention to the specific interaction and may play an important role in allocating mutual attention to those we interact with. It is argued that the synchronization of neural activities serves as a potential mechanism for tuning to significant others. Accordingly, neural synchrony marks the ability of all interactive partners to share their attention with one another and thereby facilitates their ability to synchronize their movements. This requires further validation, since according to the existing literature, it is impossible to accurately infer whether neural synchrony is the precursor to interpersonal synchrony or the product of it (55).”

*Entire paragraphs appear in the manuscript without reference, although the Authors reconstruct the debate, e.g., lines 93-99. This is unacceptable in the review article.

Response:

We thank you for this comment. Relevant references were added to this paragraph (now in lines 137 -143) and other paragraphs (as in lines 217- 236)

*Since in the paragraph "Dopamine and the neural mechanism that govern Error Detection", the authors initially introduce data on dACC, why they do not show to what part of ACC the subsequent results concern?

Response:

We now clarify that within the literature of social cognition, the dACC was found to be more associated with error monitoring than distress. The specific roles of dACC and vACC are now discussed (lines 204-217).

“It should be noted that while the dorsal part of the ACC (dACC) was found to be sensitive to expectancy violation, the ventral part of the ACC (vACC) was found to be sensitive to social distress and social rejection (88). Indeed, it has been proposed that the dACC is more involved in error monitoring than distress, and thus may be more relevant to our model. Notably, this region seems to play a major role in comparing the value of possible actions and determining what would be more beneficial: maintain the same behavior or change it (89–92). Given the involvement of dopamine in planning our motor actions by modulating the value of each possible action, it is reasonable to suggest that it may also determine the value of possible actions when the goal is to align our movements with others. In support of this notion, Gvirts and Perlmutter (89, p.5) proposed that dopamine not only “facilitates the ability of tuning-in to a specific social interaction while tuning out other interactions, but actually regulates the reward value of each of the potential interacting partners, thereby prioritizing and determining the amount of attention that should be allocated to each of them”. However, so far the role of the dopaminergic system in updating the social value associated with the interaction with a specific partner/s is yet to be determined.“

*Figure 1 (and beyond) is of very low quality and is prepared in a non-professional program, possibly a basic shape from Word. Review articles must have professional figures, I cannot imagine publishing this work with such infantile figures. Please introduce panels to all figures, prepare them in high resolution and add extensive figure captions.

Response:

We have now prepared the figures in a professional software, and added more captions.

Why do the Authors use capital letters for "Social" on line 257

Response:

This was a mistake, and is now fixed

Considerations do not end with a wise, decent discussion, compilation, forecasts and interpretations. Why is there no Discussion? This work requires a thorough mind and elaboration of a deep discussion, with reference to the most recent works. In the works cited in the manuscript, I do not find a single one from 2020, and most of them are over 10 years old.

Response:

Thank you for this constructive remark. Following the reviewer’s comment, the discussion was elaborated. We now highlight the potential role that dopamine plays in mediating both positive and negative consequences of interpersonal synchrony. We discuss the limitations of our model, mainly the fact that it does not provide an explanation for the alterations between moving in and out of synchrony that characterize social interactions. Finally, we call for future studies to investigate the proposed role of dopamine within the suggested model by methods of increasing dopamine (e.g. by methylphenidate intake) in a variety of contextual social cues.  

Round 2

Reviewer 2 Report

Dear Authors,

Thank you for a perfect revision. I have no comments and I fully recommend the publication of your work.

All the best,